# Correlation between Body Composition and Inter-Examiner Errors for Assessing Lumbar Multifidus Muscle Size, Shape and Quality Metrics with Ultrasound Imaging

**DOI:** 10.3390/bioengineering10020133

**Published:** 2023-01-18

**Authors:** Umut Varol, Elena Sánchez-Jiménez, Emma Alyette Adélaïde Leloup, Marcos José Navarro-Santana, César Fernández-de-las-Peñas, Sandra Sánchez-Jorge, Juan Antonio Valera-Calero

**Affiliations:** 1Escuela Internacional de Doctorado, Universidad Rey Juan Carlos, 29222 Alcorcón, Spain; 2Faculty of Health, Universidad Católica de Ávila, C/Canteros, s/n, 05005 Ávila, Spain; 3VALTRADOFI Research Group, Universidad Camilo José Cela, 28692 Villanueva de la Cañada, Spain; 4Department of Radiology, Rehabilitation and Physiotherapy, Complutense University of Madrid, 28223 Madrid, Spain; 5Cátedra Institucional en Docencia, Clínica e Investigación en Fisioterapia: Terapia Manual, Punción Seca y Ejercicio Terapéutico, Universidad Rey Juan Carlos, 28922 Alcorcón, Spain; 6Department of Physical Therapy, Occupational Therapy, Rehabilitation and Physical Medicine, Universidad Rey Juan Carlos, 28922 Alcorcón, Spain; 7Faculty of Health Sciences, Universidad Francisco de Vitoria, Pozuelo de Alarcón, 28223 Madrid, Spain

**Keywords:** diagnostic accuracy study, muscle composition, reliability, spine, ultrasound imaging

## Abstract

Ultrasound imaging (US) is widely used in several healthcare disciplines (including physiotherapy) for assessing multiple muscle metrics such as muscle morphology and quality. Since measuring instruments are required to demonstrate their reliability, accuracy, sensitivity, and specificity prior to their use in clinical and research settings, identifying factors affecting their diagnostic accuracy is essential. Since previous studies analyzed the impact of sociodemographic but not body composition characteristics in US errors, this study aimed to assess whether body composition metrics are correlated with ultrasound measurement errors. B-mode images of the lumbar multifidus muscle at the fifth lumbar vertebral level (L5) were acquired and analyzed in 49 healthy volunteers by two examiners (one experienced and one novel). Cross-sectional area, muscle perimeter and mean echo intensity were calculated bilaterally. A multivariate correlation matrix was calculated for assessing the inter-examiner differences with body composition metrics. Results demonstrated excellent reliability (intraclass correlation coefficient, ICC > 0.9) for assessing the muscle cross-sectional area and perimeter, and good reliability for assessing the muscle shape and mean echo intensity (ICC > 0.7). Inter-examiner errors for estimating muscle size were correlated with participants’ age (*p* value, *p* < 0.01), weight (*p* < 0.05), total and trunk lean mass (both, *p* < 0.01) and water volume (*p* < 0.05). Greater shape descriptors and mean brightness disagreements were correlated with older ages (*p* < 0.05) and total lean mass (*p* < 0.05). No correlations between age and body composition metrics were found (*p* > 0.05). This study found US to be a reliable tool for assessing muscle size, shape and mean brightness. Although aging showed no correlations with body composition changes in this sample, it was the main factor correlated with US measurement errors.

## 1. Introduction

There is an increasing use of measuring instruments among physiotherapists (PTs) as a result of the increasing demand seeking tools with acceptable sensitivity, specificity, reliability and validity to be used in different fields either instead or as a complement to manual explorations [1]. Although this collective currently uses several instruments, ultrasound imaging (US) is one of the most popular tools since is a portable, safe (since is based in ultrasound physics there is no ionizing radiation, allowing as many exams as needed in populations or situations where other imaging methods should be limited such as pregnancy), accessible, low-cost and provides real-time information [2]. 

For these reasons, PTs use US for multiple purposes. In education, several programs and universities use US as a supportive tool for teaching anatomy to first-year undergraduate students [3,4]. Regarding clinical purposes, PTs use this tool for enhancing the patients’ safety during invasive procedures by guiding the needle (either avoiding high-risk structures or for accurately reaching a specific targeted structure), as a visual feedback tool for guiding motor control exercises and for evaluating and monitoring morphological and functional changes in musculoskeletal structures [5,6,7]. In fact, recent studies used US for developing predictive models which assist in the optimal needle length recommended for interventions targeting structures with a high risk of adverse events in those cases where US is not accessible [8,9,10]. Finally, the use of US by PTs for research purposes significantly increased during the last years. In fact, after conducting a quick search in PubMed using the sentence “Ultrasound Imaging AND Physiotherapy”, the histogram shows how the number of articles published increased from a total of 66 articles between 1991 and 2015 to 207 between 2016 and 2022 (with 52 matches found in 2022 at 21 November 2022). 

This increase could be attributable to the development of US technology, enabling the evaluation of multiple objective parameters for reporting descriptive characteristics in epidemiologic studies [11,12], identifying discriminative factors [13,14] and monitoring changes in clinical trials [15,16]. For instance, Doppler is currently used for assessing vascular flows with diagnostic purposes [17] and for monitoring changes after specific interventions [18], M-mode is used for assessing muscle thickness changes during isometric contractions compared with rest thickness [19], shear-wave and strain elastography allow the examination of tissue’s stiffness [20,21] and panoramic US overcomes the difficulties of measuring muscles with large sizes which cannot be fully observed with B-mode [22,23]. In addition to the variety of US modes, the use of offline software also allows the modification of DICOM images and their measurement, providing information about the tissues’ histological and morphological characteristics [24,25].

However, one essential step before using any measuring instrument is to conduct studies analyzing the utility of these tools (e.g., validity, reliability, sensitivity and specificity). Although several studies were conducted assessing the reliability of US for assessing chronic musculoskeletal spinal pain [26,27,28,29,30], most of the evidence limited their contribution to quantifying the reliability estimates but no analyses were conducted for identifying those factors explaining whether they contribute to the measurement error variance (most of the studies focused on the examiners’ experience) [31]. Therefore, the aim of this study was to expand the list of potential contributors decreasing the reliability of US measurements by analyzing the correlation between demographic and body composition characteristics with US measurement errors of deep spinal muscles as are commonly targeted in chronic musculoskeletal spinal pain syndromes. 

## 2. Materials and Methods

### 2.1. Study Design

A cross-sectional observational study with a diagnostic accuracy study design was conducted to quantify whether demographic (i.e., age, height, sex, weight and body mass index) and body composition (i.e., percentage of total corporal fat, percentage of total corporal lean mass, total water mass and body impedance) factors could individually contribute to the inter-rater US measurement errors of lumbar multifidus muscles (i.e., muscle morphology: cross-sectional area, perimeter and shape descriptors; muscle quality: mean echo intensity). In order to enhance the presentation quality of this study, the Standards for the Reporting of Diagnostic Accuracy Studies (STARD) guidelines and checklist was followed [32].

### 2.2. Participants

The sample was recruited from a private university located in Madrid (Spain). During May 2022 and September 2022, local announcements were posted explaining the aim and procedures of the study. The data collection started in September 2022 and finished in November 2022. The eligibility criteria established were to be aged between 18 and 65 years old with no history of spinal pain during the previous year. No limitations of height or weight were set in order to recruit the most heterogeneous sample possible. Exclusion criteria were the use of any pharmacological treatment affecting muscle tone, a history of spine surgery, spinal radiculopathy or myelopathy, confirmation of severe degenerative changes or history of any musculoskeletal or medical condition (either local, such as whiplash, spondylolysis, spondylolisthesis, tumors or fractures, or widespread, such as fibromyalgia). Once eligibility criteria were verified, participants had to read and sign an informed written consent to be included in the data collection.

### 2.3. Sample Size

The minimum sample size calculation required for studies analyzing multivariate correlations was calculated based on Harry’s formula [33] (n = 50 + number of variables) since demonstrated to be a valid rule with enough power for detecting associations and factor analyses [34]. Since we included 16 potential variables, this study required at least 66 data points to be considered acceptable (n = 50 + 16 = 66). Since both sides were measured as no significant side-to-side asymmetries were expected in healthy participants, according to previous studies [13,23], 33 participants needed to be included in this study.

### 2.4. Data Collection

#### 2.4.1. Sociodemographic and Body Composition Data

A marketed bioimpedance device InBody 270 (Biospace, Urbandale, IA, USA) was used for assessing the participants’ body composition as shown in Figure 1. This multi-frequency system allows the measurement of total water volume (L), weight (kg), body fat (kg and %), body mass index (BMI = $\frac{weight\;\left(\mathrm{kg}\right)}{height\;{\left(m\right)}^{2}\;}$) [35] and muscle fat (kg and %) based on the height without footwear (m), age (years) and sex (male/female) of the participants. 

This device was used since a previous study reported an almost perfect correlation with a gold standard method (dual-energy X-ray absorptiometry, Pearson’s correlation coefficient >0.97) and excellent reliability (intraclass correlation coefficient >0.98) [36]. All measurements were carried out by an independent investigator between 9:00–11:00 a.m., weighing all participants in light clothing following the protocol described by the authors of the validity study [36].

#### 2.4.2. Examiners

All US measurements were performed by two examiners, one experienced (+10 years of experience in the use of musculoskeletal assessments) and one novice (less than one year of experience using US). The rationale for assessing only inter-examiner reliability between a novel and an experienced examiner was based on the worse reliability estimates reported in the literature (involving greater errors) compared with other conditions (intra-examiner reliability or inter-examiner reliability between two experienced examiners) [37,38,39]. 

As conducted in previous studies [27,37], the experienced examiner trained the novice on the procedures (patients’ positioning, transducer placement, US settings…) in 10 h of training distributed in two mostly practical sessions. Before starting the study, the experienced examiner ensured that the novice one acquired the skills to perform the measurements by conducting the full protocol satisfactorily.

#### 2.4.3. Ultrasound Imaging Acquisition

All ultrasound images were acquired with an Alpinion eCube i8 device and a curvilinear transducer C1-6CT (Alpinion Medical systems Co, Ltd., Gyeonggi-do, Korea). The console settings were also standard for all the acquisitions (Frequency 4 MHz, Gain 45, Depth 6 cm, Brightness 83 and Dynamic Range 72).

Immediately after conducting the body composition analyses, participants were placed in the prone position minimizing their lumbar lordosis by using a pillow under their abdomen and asked to relax their paraspinal musculature during the procedure for minimizing muscle changes due to muscle contraction [37]. 

After administering acoustic coupling gel on the lumbosacral region, the transducer was placed transversally over the sacrum. Then, the transducer was glided cranially until locating the spinous process of the fifth lumbar vertebra (L5). Once this osseous reference was visualized in the center of the image, the transducer was glided laterally to focus the lumbar multifidus muscle in the center of the image. This maneuver was performed since central areas of the images seem to improve the measurements accuracy, reliability and stability compared with lateral areas [20]. This procedure was conducted for both the left and the right lumbar multifidus once by each examiner in a randomized order (for side and examiner). Only the examiner acquiring the images was allowed to be in the room for ensuring a blinded process. 

#### 2.4.4. Measurement of Muscle Morphology and Quality

An independent researcher codified, saved and, after exporting the images acquired to DICOM format, sent the files to the examiners. Each examiner measured the images acquired by themselves in a randomized order. For ensuring the blinding, no information was shared between the examiners during this process.

All images were analyzed using the ImageJ offline DICOM software (National Institute of Health, Bethesda, MD, USA, v.1.53a). After transforming the image to a 32-bit image (which is a 256 grayscale image), the lumbar multifidus was contoured avoiding the inclusion of bone or surrounding fascia as shown in Figure 2. Finally, muscle morphology (cross-sectional area in mm^2^ and perimeter in mm), shape descriptors (circularity was calculated as 4π × Area/perimeter^2^—values range from 0 to 1, where a value of 1 indicates a perfect circle, aspect ratio was calculated as the division between the major axis and the minor axis and roundness was calculated as 4 × Area/(π × major axis^2^)) and quality (mean echo intensity calculated as the mean average brightness in this 256 grayscale within the region of interest contoured) metrics were automatically calculated. 

### 2.5. Statistical Analysis

All analyses were conducted in the Statistical Package for the Social Sciences (SPSS v.27, Armonk, NY, USA) for Mac OS, setting the significance level at *p* < 0.05 for all the analyses. Firstly, data distribution was verified using histograms and Shapiro–Wilk tests for continuous variables. *p* values < 0.05 were considered as non-normally distributed and *p* > 0.05 as normally distributed [40]. 

Secondly, descriptive statistics were used for reporting the total sample’s characteristics. Categorical data were reported as frequency and percentage for each category (e.g., number and percentage of women and men). Continuous variables were reported using central tendency metrics (i.e., mean for normal variables and median for non-normal variables) and dispersion metrics (i.e., standard deviation for normal variables and interquartile range for non-normal variables). Additionally, sociodemographic and body composition characteristics were independently reported for men and women while muscle morphology and quality characteristics were reported by gender and side. Between-group differences were analyzed using Student’s *t*-tests for independent samples, reporting the mean difference with a 95% confidence interval and considering a *p* value < 0.05 as statistically significant. 

The inter-examiner reliability analysis consisted of (1) central tendency and dispersion for each metric obtained by each examiner, (2) absolute error between examiners (absolute error was calculated since signs could underestimate the disagreement magnitude), (3) intraclass correlation coefficients (ICC_3,2_ calculated with a 2-way mixed model, consistency type), (4) standard error of measurement (SEM = Standard Deviation of the mean average × √1−ICC) and (5) minimal detectable changes (MDC = 1.96 × √2 × SEM) [41].

For assessing the correlation between the sociodemographic and body composition characteristics with US errors, a Pearson’s correlation matrix was calculated. Pearson’s correlation coefficients (r) were used to analyze the direction and strength of these correlations [42]. 

## 3. Results

From a total of 56 volunteers willing to participate in this study, 7 were excluded as they reported episodes clinically relevant to low back pain (n = 7). Therefore, since 49 asymptomatic volunteers were finally included in the data collection and analyzed, 98 lumbar multifidus muscles were studied (2 per subject, left and right). 

Table 1 summarizes the body composition characteristics of the sample (and compared by gender) and the US characteristics of the lumbar multifidus muscle (reported by gender and side). Despite males and females having a comparable age and BMI (both, *p* > 0.05), significant body composition differences were found. In general, males were taller (*p* < 0.001), heavier (*p* < 0.001) and showed greater water volume (*p* < 0.001) and lean mass (total and trunk, *p* < 0.001), whereas females showed a greater fat mass (total *p* < 0.001; trunk *p* < 0.01). Additionally, fat and lean mass percentages showed statistically significant gender differences (both for total and trunk, *p* < 0.001).

Regarding the lumbar multifidus US characteristics, no side-to-side differences were found for the metrics analyzed (all, *p* > 0.05). Although no gender differences were found for shape descriptors (all, *p* > 0.05), males exhibited larger muscle sizes (cross-sectional area and perimeter, *p* < 0.05) and lower mean echo- intensity (*p* < 0.001) compared with females. 

Table 2 shows inter-examiner reliability data. Results showed excellent ICC estimates for measuring muscle size (cross-sectional area ICC = 0.958 and muscle perimeter ICC = 0.963), good for measuring muscle quality (mean echo intensity ICC = 0.873) and good for measuring muscle shape (circularity ICC = 0.716, AR ICC = 0.710, roundness ICC = 0.707 and solidity ICC = 0.767). Indicative MDC values are also detailed for orientating whether changes in future research with longitudinal designs assessing the effect of specific interventions on these metrics are attributable to real changes (if changes are greater than MDCs) or measurement errors (if changes are smaller than MDC).

Table 3 describes the correlation between body composition characteristics with inter-examiner errors for each lumbar multifidus metric. Greater size errors were correlated with older ages (*p* < 0.01), greater weight (*p* < 0.05), greater total and trunk lean mass (both, *p* < 0.01), and lower water volume (*p* < 0.05). Greater inter-examiner errors for assessing shape descriptors were correlated with older ages (*p* < 0.05) and total lean mass (*p* < 0.05). Finally, greater mean brightness errors were correlated with older ages (*p* < 0.05) and total and trunk lean mass (*p* < 0.01).

## 4. Discussion

Up to the authors’ knowledge, this is the first study quantifying the contribution of body composition metrics on US errors. In general, we found excellent inter-examiner reliability estimates for assessing the lumbar multifidus muscle size (ICC > 0.9) and good estimates for assessing muscle shape and brightness (ICC > 0.7). Although no correlations between body fat with US errors were found, this study found lean mass and age to be the most important factors correlated with inter-examiner disagreement. 

It is highly important to understand the physics principles of US imaging for interpreting correctly the images and for being aware of their limitations [43,44,45,46]. In contrast with X-ray radiations, sound waves are non-ionizing mechanical longitudinal waves produced with a transducer at a rate of 2 MHz (1 MHz is one million cycles or pressure change produced by the sound per second) to 18 MHz in most of the US devices. This sound is produced by the vibration of certain materials when exposed to an electric current (this phenomenon is called the piezoelectric effect) [47]. 

Ultrasonography is an imaging method based on the high-frequency sound waves reflected from the tissues to the transducer [43,44,45,46]. However, not all the sound emitted by the transducer is reflected. Part of the sound is lost due to its interaction with the tissues (producing a thermic response related to the vibration) and due to refraction phenomena (changes in the direction of the sound during the interaction with some tissues) [44]. Therefore, if deeper structures receive attenuated sound (due to this sound loss), the intensity of echo is also attenuated. 

Although depth is one of the main limitations of ultrasonography, operators can modulate different US settings [48,49,50]. For instance, as a result of reducing the frequency, the sound interacts less with the tissues and, since there is less attenuation, the visualization of deeper structures significantly improves. For this reason, curvilinear transducers where the field of view is wider as long as the depth increases use lower frequencies in contrast with linear transducers (characterized by a rectangular field of view and higher frequencies for assessing structures located more superficially) [50]. Regarding this correlation between sound attenuation and frequency, Aldrich [44] provides an example illustrating how a 3 MHz wave is 50% attenuated in the first centimeter while a 6 MHz wave is 75% attenuated at the same depth. However, fewer interactions with the tissues also involve lower image quality [44]. 

Additionally, dynamic range (the ratio between the largest and smallest brightness values), gain (displacement to whiter or darker brightness) and harmonics (i.e., in contrast with lower frequencies, the use of a component frequency of a fundamental wave significantly increases the interaction with the tissues, improving the visualization of superficial structures) are also commonly used for enhancing the image quality [48,49,50,51]. 

Although the sound reflection depends on the pulse amplitude, this is not the only factor. The US reflection occurs at the boundary between two tissues with different acoustic impedance (a property associated with the density and propagation speed of the sound) [44]. If two tissues have the same acoustic impedance, their boundary will not produce any reflection (e.g., corporal fluids) and therefore the image visualized will be anechoic (black). In contrast, if the difference is very large, the US will be totally reflected (e.g., bone surfaces and air) with no further sound penetration, visualizing a hyperechoic image (white) with an acoustic posterior shadow. In between, most of the soft tissues show small to moderate acoustic impedance differences, resulting in isoechoic images with small differences in the grayscale (e.g., muscles, tendons and nerves) [44].

Since soft tissues show small acoustic impedance differences (and therefore small brightness differences in the gray scale), body composition plays a relevant role in US imaging visualization. A previous study demonstrated how age was the most determinant factor explaining the inter-examiner disagreement during US measurements of cross-sectional area, mean echo intensity and fatty infiltration estimation in deep neck muscles [31]. However, aging is a factor clearly associated with changes in body composition such as body fat gain and muscle loss [52]. In fact, Al-Sofiani et al. [53] reported that in the general population, aging is associated with a slow weight loss that becomes faster after the 75s, an annual fat mass gain of 0.40% after the 45s approximately and a significant lean mass loss (especially in men and in the leg muscles). Since these changes associated with aging could be the real reason behind the measurement disagreements, this study aimed to include participants with different ages and body composition metrics for isolating which factors really contribute to measurement errors. 

Although we initially hypothesized body fat to be an important contributor to US errors (since increased body fat increases the region of interest’s depth and sound attenuation may induce greater errors associated with visualization difficulties), the obtained results demonstrated the absence of correlation between body fat and US errors. Since correlations between age and body composition metrics were not found, this fact may explain these results. However, even if the participants’ age range was not wide enough to find correlations with body composition, age was the main factor correlated with measurement errors in agreement with a previous study [31].

Finally, it should be noted that reliability and validity studies targeting paraspinal muscles and conducted in clinical populations [26,27,31] showed poorer statistical estimates following the same procedures than those obtained for assessing healthy volunteers [28,29,30]. Although specific diagnostic accuracy for clinical populations improved significantly in recent years due to technological advances [31], histological changes correlated with chronic pain conditions such as fiber-type changes [54,55], muscle mass loss [56] and greater intra-muscular fatty infiltration [57,58,59] could also potentially be determinant factors influencing the region of interest contouring [60]. 

### Limitations

This study presents some important limitations that should be recognized. First, although we targeted wide body composition and sociodemographic ranges, future studies should include larger sample sizes with wider ranges to corroborate these findings. Secondly, we limited our recruiting strategy to healthy volunteers without chronic pain conditions. Although this strategy was useful to isolate body composition and sociodemographic factors correlated with measurement errors, further research is needed to quantify the contribution of clinical severity (in terms of disability, pain intensity, duration of symptoms, pain extent…) to measurement errors. Thus, only two examiners and a single US device were involved in the study. Further research is needed to analyze how other factors related to the examiners (e.g., time of exam, pressure with the transducer over the skin and years of experience), number of records, environmental conditions and US settings and devices contribute to the measurement errors. Finally, we only assessed the lumbar multifidus muscle at a specific level. In future studies, other musculoskeletal structures should be tested. 

## 5. Conclusions

This study found an excellent inter-examiner reliability study for assessing the lumbar multifidus muscle size and good agreement for assessing muscle shape and brightness at the L5 level. Results showed age to be one of the most important contributors to US measurement errors as our results showed significant positive correlations between participants’ age and cross-sectional area, circularity, aspect ratio, roundness and mean echo intensity errors. Additionally, less water volume was found to be correlated with greater cross-sectional area errors and greater lean mass was correlated with greater cross-sectional area, circularity and mean echo intensity measurement errors.

## Figures and Tables

**Figure 1 bioengineering-10-00133-f001:**
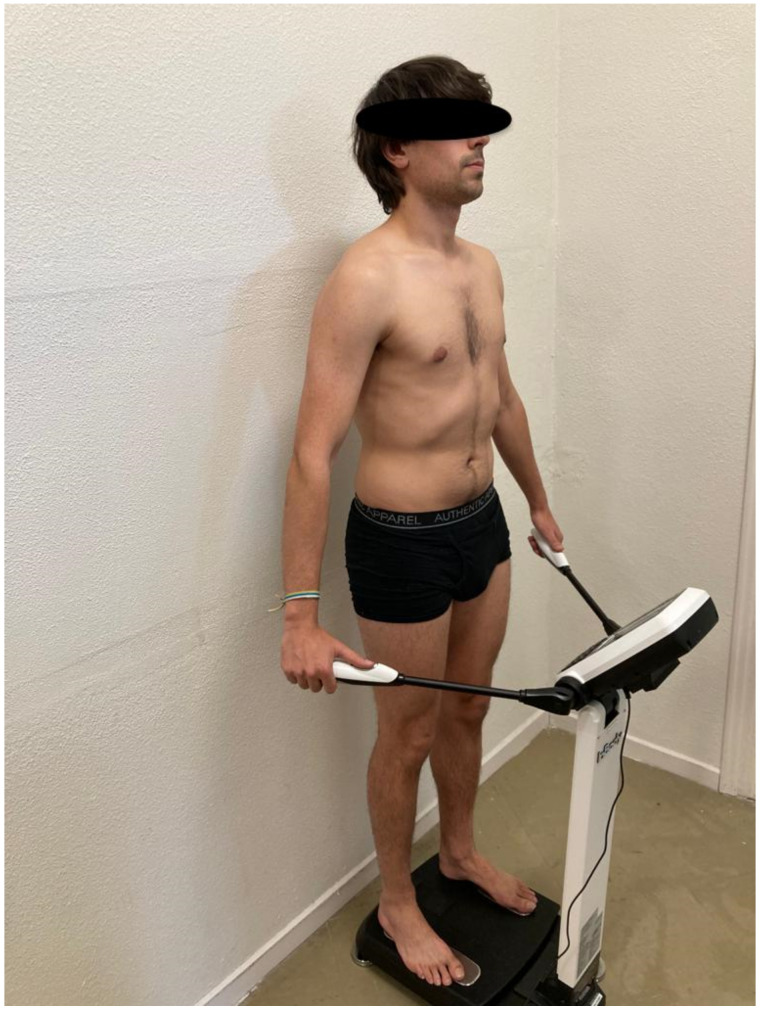
Body composition measurement using the bioimpedance device.

**Figure 2 bioengineering-10-00133-f002:**
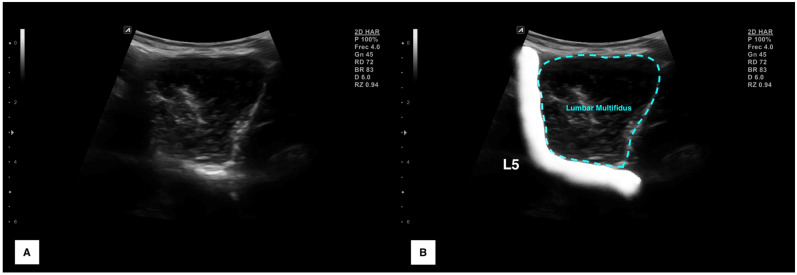
Ultrasound imaging acquisition and measurement: Raw image of the lumbar multifidus muscle acquisition at L5 (**A**) and structures identification with muscle contouring (**B**).

**Table 1 bioengineering-10-00133-t001:** Participants’ sociodemographic and US characteristics.

Variables	Total Sample (n = 49)	Gender		Side	
		Male (n = 24)	Female (n = 25)	Left (n = 48)	Right (n = 50)
*Body Composition Characteristics*					
Age (y)	22.0 ± 6.1	23.2 ± 7.0	20.1 ± 4.9	-	-
Height (m) *	1.72 ± 0.08	1.79 ± 0.05	1.67 ± 0.06	-	-
Weight (kg) *	71.6 ± 13.7	78.8 ± 10.3	65.0 ± 13.2	-	-
Body mass index (kg/m^2^)	24.0 ± 4.4	24.7 ± 3.8	23.4 ± 4.9	-	-
Fat mass					
*Total mass* (kg) *	17.3 ± 9.5	14.2 ± 8.5	20.1 ± 9.6	-	-
*Total percentage* (%) *	23.7 ± 10.3	17.3 ± 8.2	29.6 ± 8.3	-	-
*Trunk mass* (kg) **	8.7 ± 5.2	7.4 ± 4.8	10.0 ± 5.2	-	-
*Trunk percentage* (%) *	11.8 ± 5.6	8.9 ± 4.8	14.5 ± 4.9	-	-
Lean Mass					
*Total mass* (kg) *	41.0 ± 17.0	49.7 ± 17.6	33.1 ± 11.8	-	-
*Total percentage* (%) *	57.0 ± 19.8	63.2 ± 21.2	51.3 ± 16.6	-	-
*Trunk mass* (kg) *	26.9 ± 6.9	31.7 ± 6.1	22.4 ± 4.2	-	-
*Trunk percentage* (%) ***	37.6 ± 6.9	40.5 ± 7.0	34.9 ± 5.6	-	-
Water volume (L) *	40.0 ± 8.3	47.2 ± 4.8	33.4 ± 4.4	-	-
*Lumbar Multifidus Ultrasound Characteristics*					
Cross-sectional area (cm^2^) **	5.2 ± 3.7	6.4 ± 4.3	4.4 ± 2.8	5.5 ± 4.0	5.6 ± 4.3
Muscle perimeter (cm) **	8.6 ± 3.4	9.6 ± 3.9	7.8 ± 2.6	8.5 ± 3.3	8.5 ± 3.4
Circularity (0–1)	0.84 ± 0.05	0.83 ± 0.05	0.84 ± 0.05	0.83 ± 0.04	0.84 ± 0.05
Aspect ratio	1.46 ± 0.22	1.44 ± 0.25	1.48 ± 0.22	1.46 ± 0.25	1.46 ± 0.22
Roundness	0.70 ± 0.10	0.71 ± 0.11	0.68 ± 0.09	0.70 ± 0.11	0.70 ± 0.10
Solidity	0.98 ± 0.02	0.98 ± 0.02	0.99 ± 0.01	0.99 ± 0.02	0.99 ± 0.02
Mean echo intensity (0–255) *	44.9 ± 10.8	40.9 ± 8.7	48.2 ± 11.6	44.7 ± 11.1	45.1 ± 10.7

* Statistically significant gender differences (*p* < 0.001); ** Statistically significant gender differences (*p* < 0.05).

**Table 2 bioengineering-10-00133-t002:** Inter-examiner reliability for the anterior scalene US metrics.

Variables	Experienced Examiner	Novel Examiner	Absolute Error	ICC_3,2_ (95% CI)	SEM	MDC_95_
Cross-sectional area (cm^2^)	5.0 ± 3.5	5.4 ± 4.0	1.1 ± 1.1	0.958 (0.934; 0.973)	0.7	2.0
Muscle Perimeter (cm)	8.6 ± 3.4	8.6 ± 3.5	0.9 ± 0.9	0.963 (0.942; 0.976)	0.6	1.8
Circularity (0–1)	0.84 ± 0.05	0.84 ± 0.06	0.04 ± 0.03	0.716 (0.560; 0.817)	0.03	0.07
Aspect Ratio	1.49 ± 0.27	1.42 ± 0.24	0.19 ± 0.16	0.710 (0.550; 0.813)	0.15	0.40
Roundness	0.69 ± 0.12	0.72 ± 0.11	0.09 ± 0.07	0.707 (0.546; 0.811)	0.06	0.18
Solidity	0.99 ± 0.01	0.98 ± 0.02	0.01 ± 0.01	0.767 (0.639; 0.850)	0.00	0.01
Mean echo intensity (0–255)	45.6 ± 12.0	44.2 ± 11.0	5.8 ± 5.2	0.873 (0.802; 0.918)	4.3	11.8

SEM and MDC_95_ are expressed in the units described for each parameter.

**Table 3 bioengineering-10-00133-t003:** Pearson-product moment correlation matrix.

	1	2	3	4	5	6	7	8	9	10	11	12	13	14	15
1. Age															
2. Height	0.155 *														
3. Weight	0.208 **	0.519 **													
4. Body mass index	n.s.	n.s.	0.833 **												
5. Total fat mass	n.s.	−0.330 **	0.530 **	0.829 **											
6. Trunk fat mass	n.s.	n.s.	0.307 **	0.558 **	0.495 **										
7. Total lean mass	n.s.	0.284 **	0.171 *	0.409 **	n.s.	0.747 **									
8. Trunk lean mass	n.s.	0.290 **	0.228 **	n.s.	n.s.	0.643 **	0.880 **								
9. Water volume	n.s.	0.207 **	n.s.	0.458 **	n.s.	0.803 **	0.966 **	0.864 **							
10. Cross-sectional area error	0.390 **	n.s.	0.250 *	n.s.	n.s.	n.s.	0.411 **	0.404 **	−0.285 *						
11. Muscle perimeter error	n.s.	n.s.	n.s.	n.s.	n.s.	n.s.	n.s.	n.s.	n.s.	n.s.					
12. Circularity error	0.221 *	n.s.	n.s.	n.s.	n.s.	n.s.	0.271 *	n.s.	n.s.	0.677 **	n.s.				
13. Aspect Ratio error	0.243 *	n.s.	n.s.	n.s.	n.s.	n.s.	n.s.	n.s.	n.s.	n.s.	n.s.	n.s.			
14. Roundness error	0.336 **	n.s.	n.s.	n.s.	n.s.	n.s.	n.s.	n.s.	n.s.	n.s.	n.s.	n.s.	0.348 **		
15. Solidity error	n.s.	n.s.	n.s.	n.s.	n.s.	n.s.	n.s.	n.s.	n.s.	n.s.	n.s.	0.223 *	0.359 **	0.872 **	
16. Mean echo intensity error	0.276 *	n.s.	n.s.	n.s.	n.s.	n.s.	0.422 **	0.314 **	n.s.	0.221 *	0.252 *	0.250 *	0.357 **	n.s.	n.s.

Abbreviatures: n.s. non-significant. * *p* < 0.05; ** *p* < 0.01.

## Data Availability

Not applicable.

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
