# Peer review of "Correlation between Body Composition and Inter-Examiner Errors for Assessing Lumbar Multifidus Muscle Size, Shape and Quality Metrics with Ultrasound Imaging"

_bioengineering, 2023, doi:10.3390/bioengineering10020133_

Round 1

Reviewer 1 Report

Comments of this reviewer on the manuscript Bioengineering-2163153 are as follows:

1.     Based on the number of self-citations, it is obvious that the Authors are experts in the field of research to which the topic of this manuscript belongs. The topic is study of the most important contributors to measurement errors made by ultrasound imaging when assessing muscle morphology and corresponding mean echo-intensities. The list of references contains a total of 59 references, of which 23 are publications belonging to the Authors of this manuscript. It can be seen from the manuscript that there are a lot of unnecessary citations and self-citations. Such a number of self-citations is not allowed and must be reduced to 5-10 per manuscript. This is obligatory.

  1. Keywords should be singular, listed alphabetically, and followed by their abbreviations if any, namely: Keywords: Diagnostic accuracy study; Muscle composition; Reliability; Spine; Ultrasound (US) imaging.”.

3.     In this manuscript, there are the following details related to the “L5 level” used by the Authors: “B-mode images of the lumbar multifidus muscle at L5 level were acquired and analyzed in 49 healthy volunteers by two examiners (one experienced and one novel).”, “Then, the transducer was glided cranially until locating the L5 spinous process.”, “L5” is shown in Figure 2, and “This study found an excellent inter-examiner reliability study for assessing the lumbar multifidus muscle size and good agreement for assessing muscle shape and brightness at L5 level.”. It seems that this manuscript lacks an appropriate definition of the “L5 level”. Not everyone knows what the “L5 level” is. Are there levels lower or higher than L5? Responses to these questions must be included in the revised manuscript. The same applies to the other parameters used in this manuscript, namely: p, ICC, r, etc.

4.     The first sentence of Abstract is as follows: “Ultrasound imaging (US) is widely used in several healthcare disciplines (including physiotherapy) for assessing multiple muscle metrics such as muscle morphology and quality.” The term “muscle quality” seems to have been taken from the field of food technology. This term is used in several places in the manuscript. This reviewer suggests replacing the term “muscle quality” or “quality” with the corresponding term “mean echo-intensity”.

5.     In addition, this reviewer suggests replacing the term “association” and the verb “associate” with the corresponding term “correlation” and the corresponding verb “correlate”, respectively. This should be done throughout the whole manuscript.

6.     There are a couple of grammatical errors in this manuscript, such as in “…and shower greater water volume…” (“shower” should be “showed”), and “This study found an excellent inter-examiner reliability study for…” (that is, study found a study…).

7.     In “(this property is called piezoelectric effect)”, the term “phenomenon” should be used instead of the term “property”.

8.     From the content of this manuscript, the following can be extracted as important: “If two tissues have the same acoustic impedance, their boundary will not produce any reflection (e.g., corporal fluids) and therefore the image visualized will be anechoic (black).”, “In contrast, if the difference is very large, US will be totally reflected (e.g., bone surfaces and air) with no further sound penetration, visualizing a hyperechoic image (white) with an acoustic posterior shadow.”, and “Although no associations between body fat with US errors were found, this study found lean mass and age to be the most important factors associated with inter-examiner disagreement.” Accordingly, the body fat and the lean mass have acoustic impedances different from the acoustic impedance of muscles studied, while the age cannot be correlated with the acoustic impedances. How is the acoustic impedance calculated? What are the acoustic impedance values of the investigated tissues, body fat, lean mass and muscles? All these responses should be included in the text of the manuscript.

9.     Along with the presentation of the results and their discussion, the values of the acoustic impedances of the tissues studied should be given, and the discussion should be rearranged from the aspect of the acoustic impedance values. Thus, the quality of this study will be raised to a higher level.

10. The conclusions should also be correlated with the acoustic impedance. This is very important because the significance of the conclusions presented is very low. The concluding section could also be expanded, while the conclusions could be quantified.

Author Response

We would like to thank the reviewers for their comments, which we believe have clarified many aspects of the manuscript. We have edited the text according to the suggestions from the reviewers. We have highlighted all changes in yellow throughout the manuscript. A point-by-point response is presented below.

Reviewer 1

Based on the number of self-citations, it is obvious that the Authors are experts in the field of research to which the topic of this manuscript belongs. The topic is study of the most important contributors to measurement errors made by ultrasound imaging when assessing muscle morphology and corresponding mean echo-intensities. The list of references contains a total of 59 references, of which 23 are publications belonging to the Authors of this manuscript. It can be seen from the manuscript that there are a lot of unnecessary citations and self-citations. Such a number of self-citations is not allowed and must be reduced to 5-10 per manuscript. This is obligatory.

Response: We revised and arranged this issue.

Keywords should be singular, listed alphabetically, and followed by their abbreviations if any, namely: Keywords: Diagnostic accuracy study; Muscle composition; Reliability; Spine; Ultrasound (US) imaging.”.

Response: Keywords were modified as suggested.

In this manuscript, there are the following details related to the “L5 level” used by the Authors: “B-mode images of the lumbar multifidus muscle at L5 level were acquired and analyzed in 49 healthy volunteers by two examiners (one experienced and one novel).”, “Then, the transducer was glided cranially until locating the L5 spinous process.”, “L5” is shown in Figure 2, and “This study found an excellent inter-examiner reliability study for assessing the lumbar multifidus muscle size and good agreement for assessing muscle shape and brightness at L5 level.”. It seems that this manuscript lacks an appropriate definition of the “L5 level”. Not everyone knows what the “L5 level” is. Are there levels lower or higher than L5? Responses to these questions must be included in the revised manuscript. The same applies to the other parameters used in this manuscript, namely: p, ICC, r, etc.

Response: L5 referred the 5th lumbar vertebra. We clarified those concepts the first time they appear.

The first sentence of Abstract is as follows: “Ultrasound imaging (US) is widely used in several healthcare disciplines (including physiotherapy) for assessing multiple muscle metrics such as muscle morphology and quality.” The term “muscle quality” seems to have been taken from the field of food technology. This term is used in several places in the manuscript. This reviewer suggests replacing the term “muscle quality” or “quality” with the corresponding term “mean echo-intensity”.

Response: Both terms are totally different, so we used each one in the most appropriate context. Muscle quality is accepted in rehabilitation and is defined as the amount of strength and/or power per unit of muscle mass while mean echo-intensity is a muscle quality metric. Muscles with a greater proportion of non-contractile tissue are brighter, accompanied by a higher mean echo-intensity value, and ultimately lower muscle quality. Mean echo-intensity is used as a metric since previous studies associated this parameter with muscle strength, power, and function (ref. 24). Since we used both terms in their appropriate context, we would prefer to keep them as is.

In addition, this reviewer suggests replacing the term “association” and the verb “associate” with the corresponding term “correlation” and the corresponding verb “correlate”, respectively. This should be done throughout the whole manuscript.

Response: These replacements were done as suggested.

There are a couple of grammatical errors in this manuscript, such as in “…and shower greater water volume…” (“shower” should be “showed”), and “This study found an excellent inter-examiner reliability study for…” (that is, study found a study…).

Response: As required by other reviewer, the manuscript was sent for English revision for misspellings and grammar correction.

In “(this property is called piezoelectric effect)”, the term “phenomenon” should be used instead of the term “property”.

Response: The term was modified as suggested.

From the content of this manuscript, the following can be extracted as important: “If two tissues have the same acoustic impedance, their boundary will not produce any reflection (e.g., corporal fluids) and therefore the image visualized will be anechoic (black).”, “In contrast, if the difference is very large, US will be totally reflected (e.g., bone surfaces and air) with no further sound penetration, visualizing a hyperechoic image (white) with an acoustic posterior shadow.”, and “Although no associations between body fat with US errors were found, this study found lean mass and age to be the most important factors associated with inter-examiner disagreement.” Accordingly, the body fat and the lean mass have acoustic impedances different from the acoustic impedance of muscles studied, while the age cannot be correlated with the acoustic impedances. How is the acoustic impedance calculated? What are the acoustic impedance values of the investigated tissues, body fat, lean mass and muscles? All these responses should be included in the text of the manuscript.

Response: This explanation was given for clarifying the physical mechanisms of US imaging, the reason why different tissues are visualized with different brightness and the relation between image brightness and acoustic impedance. In this study acoustic impedance of biological tissues was not assessed since was not relevant as we used an indirect indicator of acoustic impedance (brightness).

Along with the presentation of the results and their discussion, the values of the acoustic impedances of the tissues studied should be given, and the discussion should be rearranged from the aspect of the acoustic impedance values. Thus, the quality of this study will be raised to a higher level.

Response: As answered in the previous comment, acoustic impedance values were not calculated since we used an indirect indicator of acoustic impedance.

The conclusions should also be correlated with the acoustic impedance. This is very important because the significance of the conclusions presented is very low. The concluding section could also be expanded, while the conclusions could be quantified.

Response: Including information about acoustic impedance is not possible. Conclusion was expanded based on the results obtained.

Reviewer 2 Report

The investigation of body composition's features in ultrasound imaging is of interest. The manuscript is well drafted. Data is sufficient and the analysis method is accurate. The experiment is well-designed and the conclusions are solid. 

A small minor revision can be done:

Displays of some equations are not correct. e.g. SEM in Line 225, MDC in Line 226, etc.

Author Response

We would like to thank the reviewers for their comments, which we believe have clarified many aspects of the manuscript. We have edited the text according to the suggestions from the reviewers. We have highlighted all changes in yellow throughout the manuscript. A point-by-point response is presented below.

Reviewer 2

The investigation of body composition's features in ultrasound imaging is of interest. The manuscript is well drafted. Data is sufficient and the analysis method is accurate. The experiment is well-designed and the conclusions are solid.

Response: Thank you for this positive feedback.

A small minor revision can be done:

Displays of some equations are not correct. e.g. SEM in Line 225, MDC in Line 226, etc.

Response: We revised the equations as suggested.

Reviewer 3 Report

the paper is interesting and sound in the idea ...but some comments should be addressed 

line 117-123: sample size calculation is not clear please clarify 

line 129: body mass index needs a reference

line 167: a more setting for ultrasound is needed in order to repeat the same measurement by another author 

more clear ultrasound pictures are needed 

I note that the English language in your manuscript could be improved. Your manuscript can Once you receive the comments from the reviewers, please improve the language quality. Please do not submit a revised version until then.

Author Response

We would like to thank the reviewers for their comments, which we believe have clarified many aspects of the manuscript. We have edited the text according to the suggestions from the reviewers. We have highlighted all changes in yellow throughout the manuscript. A point-by-point response is presented below.

the paper is interesting and sound in the idea ...but some comments should be addressed 

line 117-123: sample size calculation is not clear please clarify 

Response: We clarified the sample size calculation as suggested

line 129: body mass index needs a reference

Response: The following reference has been added:

35. Gallagher D, Heymsfield SB, Heo M, Jebb SA, Murgatroyd PR, Sakamoto Y. Healthy percentage body fat ranges: an approach for developing guidelines based on body mass index. Am J Clin Nutr. 2000;72(3):694-701. doi:10.1093/ajcn/72.3.694

line 167: a more setting for ultrasound is needed in order to repeat the same measurement by another author 

Response: We are sorry, but we do not understand this requirement. All US settings information was provided in lines 166-169

“All ultrasound images were acquired with a Alpinion eCube i8 device and a curvilinear transducer C1-6CT (Alpinion Medical systems Co, Ltd., Gyeonggi-do, Korea). The console settings were also standard for all the acquisitions (Frequency 4 MHz, Gain 45, Depth 6 cm, Brightness 83 and Dynamic Range 72”.

More clear ultrasound pictures are needed 

Response: We do not understand this request. The image attached as Figure 2 shows the raw US image acquired (Figure 2A) and all the structures visualized were labeled (Figure 2B).

I note that the English language in your manuscript could be improved. Your manuscript can Once you receive the comments from the reviewers, please improve the language quality. Please do not submit a revised version until then.

Response: We apologize for the grammar errors and misspellings along the manuscript. The full manuscript has been carefully revised by a native colleague.

Round 2

Reviewer 1 Report

It seems that the majority of my comments have been addressed in appropriate way. 

Reviewer 3 Report

The paper us extensively improved now